# Derivative-Free Iterative One-Step Reconstruction for Multispectral CT

**DOI:** 10.3390/jimaging10050098

**Published:** 2024-04-24

**Authors:** Thomas Prohaszka, Lukas Neumann, Markus Haltmeier

**Affiliations:** 1Institute of Basic Sciences in Engineering Science, University of Innsbruck, Technikerstrasse 13, 6020 Innsbruck, Austria; thomas.prohaszka@student.uibk.ac.at; 2Department of Mathematics, University of Innsbruck Technikerstrasse 13, 6020 Innsbruck, Austria

**Keywords:** inverse problems, coupled physics problems, multispectral CT, derivative-free iteratzions, inverse problems

## Abstract

Image reconstruction in multispectral computed tomography (MSCT) requires solving a challenging nonlinear inverse problem, commonly tackled via iterative optimization algorithms. Existing methods necessitate computing the derivative of the forward map and potentially its regularized inverse. In this work, we present a simple yet highly effective algorithm for MSCT image reconstruction, utilizing iterative update mechanisms that leverage the full forward model in the forward step and a derivative-free adjoint problem. Our approach demonstrates both fast convergence and superior performance compared to existing algorithms, making it an interesting candidate for future work. We also discuss further generalizations of our method and its combination with additional regularization and other data discrepancy terms.

## 1. Introduction

Classical computed tomography (CT) is based on the inversion of the linear Radon transform, where a scalar-valued attenuation map μ:X→R of the patient is recovered from observation of its Radon transform Rμ:L→R derived from projection data. Here and below $X\subseteq R^{d}$ is the image domain in d=2,3 dimensions, and L is the set of integration lines. While sufficient in many applications, the linear problem ignores the polychromatic nature of the X-rays and the energy-dependent absorption characteristics of real-world objects. The sample is more accurately represented by a family of attenuation maps μ(e):X→R dependent on the photon energy e∈(0,∞). Recovering a single μ from projection data using a single energy bin results in a mixture of density maps from different energies resulting in severe nonuniqueness. Additionally, the nonlinearity results in beam hardening artifacts that may be partially accounted for by iterative algorithms or analytic modeling [1,2,3,4,5,6]. In order to overcome such weaknesses, the idea of multispectral CT (MSCT) is to measure projection data for different energy bands, which are then used to reconstruct multiple attenuation maps. The reconstruction problem, however, becomes nonlinear and much more challenging than pure Radon inversion [7,8,9,10,11,12,13]. In this work, we develop a simple and efficient strategy for tackling the nonlinearity.

### 1.1. Multispectral CT

Specifically, in this work, we use the material decomposition paradigm in MSCT. In the material decomposition approach, it is assumed that the energy-dependent attenuation maps μ(e) can be written as $\mu \left(e,x\right)=\sum_{m=1}^{M}\mu_{m}\left(e\right)f_{m}\left(x\right)$, where $f_{1},f_{2},\ldots ,f_{M}:X\to R$ are the densities of *M* separate materials (with $f_{m}\left(x\right)\in \left[0,1\right]$) to be recovered and $\mu_{m}\left(e\right)$ are known and tabled absorption characteristic of the *m*-th material. The aim is to recover the material densities from data collected for several energy bands. This does not only allow to improve image quality, but also offers a broad range of applications, as it reconstructs multiple images encoding different characteristics of specific regions, which enables a deeper understanding of the object under examination. Recent significant advancements in the manufacturing of energy-sensitive sensors [14,15] has considerably increased the interest in and applicability of MSCT.

Assuming *B* spectral measurements $Y_{1},\ldots ,Y_{B}$, the material decomposition problem in MSCT can be written as the problem of recovering $f_{1},\ldots ,f_{M}$ based on the modeling equation ${\left(Y_{b}\right)}_{b=1}^{B}=\Phi \left({\left(Rf_{m}\right)}_{m=1}^{M}\right)$. Here, $Rf_{m}$ is the Radon transform applied to the *m*-th material density map $f_{m}$, and Φ is a nonlinear map. Classical CT would correspond to the unrealistic case where the application of the pointwise logarithm in the modeling equation results in a linear inverse problem. In MSCT, one accounts for the full nonlinear problem for the unknown density maps $f_{m}$.

### 1.2. Two-Step and One-Step Algorithms

Various algorithms have been developed for solving the reconstruction problem in MSCT. They can be broadly classified into two categories: two-step methods and one-step algorithms. The idea of earlier two-step methods is to perform Radon inversion and material decomposition in two separate steps. Material decomposition can be performed either in the projection domain L (before Radon inversion) or in the image domain X (after Radon inversion). Both methods have their specific advantages and disadvantages. The image-domain decomposition approach allows incorporating prior information about the objects that is naturally contained in the image domain X. However, the nonlinear nature of the problem leads to approximate linear models that introduce severe reconstruction artifacts. The projection-domain decomposition approach, on the other hand, allows working with the correct nonlinear model. However, the prior structure in the Radon domain is not directly available. See the works [12,13,16,17,18] and references therein for proposed two-step approaches.

One-step methods reconstruct the material densities $f_{1},\ldots ,f_{M}$ through iterative minimization techniques and thus overcome the drawbacks of both two-step methods. For some one-step algorithms, we refer to [7,8,11,19,20,21,22,23,24]. Despite their superior performance, such one-step iterative algorithms are computationally expensive. Existing methods require many iterative steps due to poor conditioning of the problem with computationally expensive iterative steps. The algorithms proposed in this paper are specific one-step algorithms that address these two drawbacks of existing one-step methods. The structure of our proposed algorithm is shown in Figure 1.

### 1.3. Our Contributions

The following list summarizes the main contributions of this paper:We present a novel derivative-free algorithm designed to combine the advantages of one-step and two-step approaches. To achieve this, we introduce a simple and computationally efficient iterative update that incorporates appropriate preconditioning.Image reconstruction is performed in the image domain, which naturally allows for the inclusion of an image smoothness prior. Our method can be combined with additional regularization. However, in order to show the method in its pure form, we will not include such a modification.Our methods integrate benefits of two-step approaches by separating iterative updates into two parts. Moreover, the main ingredient that makes the algorithm efficient is the use of the full nonlinear forward model but linearization around zero for the adjoint problem. In addition to avoiding computation and evaluation of the derivative of the forward map, this also allows for including simple but efficient channel preconditioning.

## 2. Mathematical Modelling of MSCT

We assume that the object to be imaged lies in some domain $X\subseteq R^{2}$ and consists of a combination of *M* different materials with densities $f_{m}:X\to R$ with m=1,…,M. Each material has a separate mass attenuation coefficient $\mu_{m}:\left[0,\infty \right)\to \left[0,\infty \right)$, which is a known function of the X-ray energy *e*. The total energy dependent (linear) X-ray attenuation coefficient is then given by:(1)$$\mu \left(x,e\right)=\sum_{m=1}^{M}\mu_{m}\left(e\right)f_{m}\left(x\right)\;\;\mathrm{for}\;\left(x,e\right)\in X\times R\;.$$
Assuming that the material specific attenuation functions $\mu_{m}\left(e\right)$ are known, the goal is to recover densities $f_{m}$ from indirect x-ray measurements using different energy bins.

### 2.1. Continuous Model

We start with continuous modeling, where the quantities involved are functions on continuous domains that will be discretized later. Suppose that X-ray energy with a known incident spectral density $I_{0}\left(e\right)$ is sent along a line ℓ∈L from the source position to the detector position. While propagating along *ℓ*, the X-rays are attenuated according to μ(x,e) defined in (Equation 1). This results in an outgoing spectral density $I_{1}\left(e\right)=I_{0}\left(e\right)\exp\left(-\int_{\ell }\mu \left(x,e\right)d\ell \left(x\right)\right)$ at the detector. The energy-sensitive detector with the spectral profile $\hat{s}\left(e\right)$ records the integral $\int_{0}^{\infty }\hat{s}\left(e\right)I_{1}\left(e\right)de$. Denoting the product of the incident spectral density of the source and the spectral sensitivity of the detector by $s\left(e\right)≜I_{0}\left(e\right)\hat{s}\left(e\right)$, referred to as the effective spectrum, the recorded data are given by:(2)$$\begin{array}{cc} Y(s,\ell )\; & =\int_{0}^{\infty }s\left(e\right)\exp\left(-\int_{\ell }\mu \left(x,e\right)d\ell \left(x\right))\right)de\\ \; & =\int_{0}^{\infty }s\left(e\right)\exp\left(-\sum_{m=1}^{M}\mu_{m}\left(e\right)\int_{\ell }f_{m}d\ell \left(x\right)\right)de\;. \end{array}$$
The data in Equation (Equation 2) represent a single measurement in MSCT. The goal of material decomposition in MSCT is to determine the density distributions $f_{m}$ from multispectral X-ray measurements obtained by varying the line *ℓ* and the effective spectra *s*.

For simplicity of presentation, we consider parallel beam mode, where any line ℓ=ℓ(θ,r) is parametrized by its normal vector θ and its distance *r* from the origin. In this case, $\int_{\ell }f_{m}d\ell \left(x\right)=Rf_{m}\left(\theta ,x\right)$ is given by the classical Radon transform of $f_{m}$. Assuming further a total number of *B* different effective spectra and writing $f=(f_{1},\ldots ,f_{M})$, we obtain the continuous MSCT forward model:(3)$$Y={\left[\int_{0}^{\infty }s_{b}\left(e\right)\exp\left(-\sum_{m=1}^{M}\mu_{m}\left(e\right)R\left(f_{m}\right)\right)de\right]}_{b=1,\ldots B}\;.$$
Equation (Equation 3) gives the complete continuous forward model in material decomposition in multispectral CT. The unknown f consists of *M* functions $f_{1},\ldots ,f_{M}$ defined on the image domain X and the data of *B* functions $Y_{1},\ldots ,Y_{B}$ defined on the projection domain $L=S^{1}\times R$. The methods that we describe, however, would also work with a three-dimensional image domain and a general projection domain L of lines in $R^{3}$.

### 2.2. Discretization

In order to avoid technical details and to concentrate on the main ideas, we derive the algorithm for the discrete forward model throughout this paper. For that purpose, we represent the material densities via a discrete column vector $X_{1},\ldots ,X_{M}\in R^{N_{x}}$ and the Radon transform via a matrix $A\in R^{N_{y}\times N_{x}}$, where $N_{x}$ is the number of discretization points in the image domain and $N_{y}$ is the number of lines used in the projection domain. Furthermore, we discretize the effective energy spectra by vectors $S_{1},\ldots ,S_{B}\in R^{E}$ and the known material attenuations by vectors $\mu_{1},\ldots ,\mu_{M}\in R^{E}$, where *E* is number of discretization points of the energy variable. The discretization of (Equation 3) yields the following discrete image reconstruction problem.

**Problem** **1 (Discrete MSCT image reconstruction problem).**
*Recover the unknown $X\in R^{N_{x}\times M}$ from data $Y=F\left(X\right)+\delta \in R^{N_{y}\times B}$ where:*

(4)
$$F\left(X\right)≜{\left(S\cdot \exp_{E\times N_{y}}\left(-M\cdot {\left(AX\right)}^{⊺}\right)\right)}^{⊺}$$

*Here and below, we use the convention that the boldface notation $\exp_{E\times N_{y}}$ indicates that the scalar function exp is applied pointwise to a vector in $R^{E\times N_{y}}$. Furthermore, in (Equation 4):*

*The columns of $X=[X_{1},\ldots ,X_{M}]$ are the discrete material images;*

*$A\in R^{N_{y}\times N_{x}}$ is the discretized Radon transform;*

*The columns of $M=\left[M_{1},\ldots ,M_{M}\right]\in R^{E\times M}$ are the discretized material attenuations;*

*The columns of $S^{⊺}=\left[S_{1}^{⊺},\ldots ,S_{B}^{⊺}\right]\in R^{E\times B}$ the discretized effective spectra;*

*The columns of $Y=[Y_{1},\ldots ,Y_{B}]$ are the observed spectral data.*



Note that we included the transpose operations in (Equation 4) such that all involved linear operations can be written as matrix multiplications from left. Alternatively, we can also write $F\left(X\right)=\exp_{N_{y}\times E}\left(-AXM^{⊺}\right)S^{⊺}$, reflecting the fact that the discrete Radon transform A operates on the columns and the matrices M and S on the rows of $X\in R^{N_{y}\times M}$. At some places, we will denote operation of M on *X* from the right by $M⋄X≜X\cdot M^{⊺}$ such that we have $F\left(X\right)=S⋄\exp_{N_{y}\times E}\left(-M⋄AX\right)$. The structure of the discrete forward model is illustrated in Figure 2.

**Remark** **1 (Recalibration).**
*With $\langle S_{b},1\rangle =\sum_{e=1}^{E}S_{b,e}$ we obtain:*

(5)
$$F{\left(0\right)}^{⊺}=S\cdot \exp_{E\times N_{y}}\left(0\right)=S\cdot 1_{E\times N_{y}}=\left(\begin{array}{ccc} \langle 1,S_{1}\rangle & \ldots & \langle 1,S_{1}\rangle \\ \vdots & \; & \vdots \\ \langle 1,S_{B}\rangle & \ldots & \langle 1,S_{B}\rangle \end{array}\right)\;.$$

*Thus, with *⊙* denoting pointwise multiplication (also known as Hadamard product) and *⊘* the pointwise division, we get:*

(6)
$$\begin{array}{c} F{\left(X\right)}^{⊺}⊘F{\left(0\right)}^{⊺}=\left(\begin{array}{ccc} 1/\langle 1,S_{1}\rangle & \ldots & 1/\langle 1,S_{1}\rangle \\ \vdots & & \vdots \\ 1/\langle S_{b},1\rangle & \ldots & 1/\langle S_{b},1\rangle \end{array}\right)⊙\left(S\cdot \exp_{E\times N_{y}}\left(-M\cdot {\left(AX\right)}^{⊺}\right)\right)\\ =\left(\begin{array}{c} S_{1}/\langle 1,S_{1}\rangle \\ \vdots \\ S_{B}/\langle 1,S_{B}\rangle \end{array}\right)\cdot \exp_{E\times N_{y}}\left(-M\cdot {\left(AX\right)}^{⊺}\right)\;. \end{array}$$

*This means that the recalibrated forward model $F{\left(X\right)}^{⊺}⊘F{\left(0\right)}^{⊺}$ is the same as the original forward model with normalized effective spectra $S_{b}/\langle S_{b},1\rangle$. The matrix with normalized spectra can be written as $S⊘(S\cdot 1_{E\times E})$.*


In this work we, use rescaled data, to which we apply the pointwise logarithm $\log_{N_{y}\times B}$ and the corresponding least squares (LSQ) functional.

**Definition** **1 (Forward model and LSQ functional).**
*We define the logarithmic forward model and the LSQ functional by:*

(7)
$$\begin{array}{cc} \; & H:R^{N_{x}\times M}\to R^{N_{y}\times B}:Y\mapsto \log_{N_{y}\times B}\left(F\left(X\right)⊘F\left(0\right)\right) \end{array}$$


(8)
$$\begin{array}{cc} \; & D:R^{N_{x}\times M}\to R:X\mapsto {∥H\left(X\right)-Y^{H}∥}_{2}^{2}/2\;. \end{array}$$

*where $F\left(X\right)=\exp_{N_{y}\times E}\left(-AXM^{⊺}\right)S^{⊺}$ is defined in (Equation 4), $Y\in R^{N_{y}\times B}$ are the given data, $Y_{H}=\log_{N_{y}\times B}\left(Y⊘F\left(0\right)\right)$ the modified data, and $\log_{N_{y}\times B}\left(\cdot \right)$ the pointwise logarithm.*


Using the notations of Definition 1, material decomposition in MSCT amounts to the near solution of $H\left(X\right)=Y^{H}$ or the near minimization of D.

**Remark** **2 (Noise modelling).**
*In the statistical context, LSQ minimization derives from maximum likelihood estimation using a Gaussian noise model on $Y_{H}$. From a statistical perspective, maximum likelihood estimation for Poisson noise on Y might be more reasonable, resulting in $L\left(X\right)=\langle F\left(X\right),1\rangle -\langle \log\left(F\left(X\right)\right),Y\rangle \to \min_{X}$. As the focus in this paper is the derivation of an efficient reconstruction algorithm rather than statistical optimality, we work with D. However, we expect that our strategy can also be applied to L instead of D.*


**Remark** **3** (Regularization).
*Due to the ill-conditioning of $H\left(X\right)=Y^{H}$, the reconstruction problem has to be regularized [25,26]. In the context of LSQ minimization, a natural approach is variational regularization, where one considers ${∥H\left(X\right)-Y^{H}∥}_{2}^{2}/2+\alpha R\left(X\right)$ instead of D with a regularization functional R(X). Recently, in [27], the plug-and-play method has been identified as a regularization technique where the regularization is incorporated by a denoiser. Another class is given by iterative regularization [28], where regularization comes from early stopping. All these regularization methods require the gradient of D, which we compute below.*


## 3. Algorithm Development

In this section, we derive the proposed algorithms for MSCT based on channel preconditioning (CP). The first one (CP-full) integrates channel preconditioning into a gradient scheme. In the second algorithm (CP-fast), the derivative of the forward map is replaced by the derivative at zero. Both methods greatly reduce the number of iterations compared to standard gradient methods and the numerical cost per iteration compared to Newton type methods. Before presenting our algorithms, we start by computing derivatives and gradients and recall existing gradient and Newton type methods.

### 3.1. Derivatives Computation

Standard algorithms for minimizing (Equation 8) require the derivative of the forward map and the gradient of the LSQ functional that we compute next. Recall the original and logarithmic MSCT forward operator $F,H:R^{N_{x}\times M}\to R^{N_{y}\times B}$ and the LSQ functional D defined by (Equation 4), (Equation 7), and (Equation 8). We equip $R^{N_{x}\times M}$ and $R^{N_{y}\times B}$ with the Hilbert space structure induced by the standard inner product $\langle \xi_{1},\xi_{2}\rangle =\sum_{i,j}\xi_{1}\left[i,j\right]\xi_{2}\left[i,j\right]$. Furthermore, we use $H^{\prime }\left[X\right]:R^{N_{x}\times M}\to R^{N_{y}\times B}$ to denote the derivative of H at location $X\in R^{N_{x}\times M}$ and $D^{\prime }\left[X\right]:R^{N_{x}\times M}\to R$ and $\nabla D\left[X\right]\in R^{N_{x}\times M}$ to denote the derivative and the gradient of D at *X*, respectively.

**Remark** **4 (Gradients, inner products, and preconditioning).**
*By the definition of gradients, we have $\langle \nabla D\left[X\right],\xi \rangle ={\left(D^{\prime }\left[X\right]\right)}^{*}\left(\xi \right)$, where ${\left(\cdot \right)}^{*}$ denotes the adjoint of a linear operator. Furthermore, by the chain rule, $\nabla D\left[X\right]={\left(H^{\prime }\left[X\right]\right)}^{\cdot }\left(H\left(X\right)-Y^{H}\right)$. Gradients and adjoints depend on the chosen inner product. For example, the inner product $\langle \xi_{1},U^{-1}\xi_{2}\rangle$ on the image space with a positive-definite matrix U yields the modified gradient U·∇D[X]. Choosing U such that U·∇D[X] has improved condition significantly improves gradient based methods for minimizing D.*


**Remark** **5 (Some calculus rules).**
*For the following computation we use some elementary calculus rules listed next. Let $G_{1},G_{2}:R^{N_{x}}\to R^{N_{y}}$ be vector valued functions and ϕ:R→R a scalar function with derivative ψ:R→R. Then, for $X,\xi \in R^{N_{x}}$, we have:*

(9)
$$\begin{array}{cc} {\left(G_{1}⊙G_{2}\right)}^{\prime }\left[X\right]\left(\xi \right) & =\left(G_{1}^{\prime }\left[X\right]\left(\xi \right)\right)⊙\left(G_{2}\left(X\right)\right)+\left(G_{1}\left(X\right)\right)⊙\left(G^{\prime }\left[X\right]\left(\xi \right)\right) \end{array}$$


(10)
$$\begin{array}{cc} \phi_{N}^{\prime }\left[X\right]\left(\xi \right) & =\psi_{N}\left(X\right)⊙\xi . \end{array}$$

*As usual, we define the vector value functions $\phi_{N},\psi_{N}:R^{N_{x}}\to R^{N_{x}}$ by pointwise application $\phi_{N}\left(X\right)={\left(\phi \left(X_{i}\right)\right)}_{i}$ and $\psi_{N}\left(X\right)={\left(\psi \left(X_{i}\right)\right)}_{i}$.*


We have the following explicit expressions for derivatives, adjoints, and gradients.

**Theorem** **1 (Derivatives computation).**
*Let $F,H:R^{N_{x}\times M}\to R^{N_{y}\times B}$ and D be defined by (Equation 4), (Equation 7) and (Equation 8). The derivative of F and H, as well as the adjoint and the gradient of D are given by:*

(11)
$$\begin{array}{cc} F^{\prime }\left[X\right]{\left(\xi \right)}^{⊺} & =-S\cdot (Q_{X}⊙\left(M{\left(A\xi \right)}^{⊺}\right) \end{array}$$


(12)
$$\begin{array}{cc} H^{\prime }\left[X\right]{\left(\xi \right)}^{⊺} & =-F{\left(X\right)}^{-T}⊙\left(S\cdot \left(Q_{X}⊙\left(M{\left(A\xi \right)}^{⊺}\right)\right)\right) \end{array}$$


(13)
$$\begin{array}{cc} F^{\prime }{\left[X\right]}^{*}\left(\eta \right) & =-A^{⊺}\cdot {\left(M^{⊺}\cdot \left(Q_{X}⊙\left(S^{⊺}\eta^{⊺}\right)\right)\right)}^{⊺} \end{array}$$


(14)
$$\begin{array}{cc} H^{\prime }{\left[X\right]}^{*}\left(\eta \right) & =-A^{⊺}\cdot {\left(M^{⊺}\cdot \left(Q_{X}⊙\left(S^{⊺}\left(F{\left(X\right)}^{-T}⊙\eta^{⊺}\right)\right)\right)\right)}^{⊺} \end{array}$$


(15)
$$\begin{array}{cc} \nabla D[X] & =-A^{⊺}\cdot {\left(M^{⊺}\cdot \left(Q_{X}⊙\left(S^{⊺}\left(F{\left(X\right)}^{-T}⊙{\left(H\left(X\right)-Y^{H}\right)}^{⊺}\right)\right)\right)\right)}^{⊺}\;, \end{array}$$

*where $Q_{X}≜\exp_{E\times N_{y}}\left(-M{\left(AX\right)}^{⊺}\right)$ denotes virtual spectrally resolved data.*


**Proof.** The proof is given in the Appendix A. □

For CP-fast, the derivative of H at zero plays a central role.

**Remark** **6 (Derivative at zero).**
*Let us consider the derivative at the zero image X=0. In this case, we have $Q_{0}:=\exp_{E\times N_{y}}\left(-M{\left(A0\right)}^{⊺}\right)=1_{E\times N_{y}}$, and therefore, $H^{\prime }\left[0\right]{\left(\xi \right)}^{⊺}=-\left(SM{\left(A\xi \right)}^{⊺}\right)⊘F{\left(0\right)}^{⊺}$ and $H^{\prime }{\left[0\right]}^{*}\left(\eta \right)=-A^{⊺}\left(\eta ⊘F\left(0\right)\right)\cdot S\cdot M$. Using that $F{\left(0\right)}^{⊺}=S\cdot 1_{E\times N_{y}}$, we get:*

(16)
$$\begin{array}{cc} H^{\prime }\left[0\right]{\left(\xi \right)}^{⊺} & =-U{\left(A\xi \right)}^{⊺} \end{array}$$


(17)
$$\begin{array}{cc} H^{\prime }{\left[0\right]}^{*}\left(\eta \right) & =-A^{⊺}\eta^{⊺}U \end{array}$$


(18)
$$\begin{array}{cc} U & ≜\left(SM\right)⊘\left(S\cdot 1_{E\times M}\right)\;. \end{array}$$

*The derivative $H^{\prime }\left[0\right]\left(\xi \right)=-A\xi U^{⊺}$ may also be seen as the linearization of H around zero. It has been used previously in MSCT and can be simply derived by first order Taylor series approximation [12]. In fact, with $F{\left(0\right)}_{\ell ,b}=\langle S_{b},1_{E\times 1}\rangle$, we find:*

(19)
$$\begin{array}{cc} \log\left(\frac{\langle S_{b},\exp\left(-M{\left(A\xi \right)}_{\ell }^{⊺}\right)\rangle }{\langle S_{b},1_{E\times 1}\rangle }\right) & ≃\log\left(\frac{\langle S_{b},\left(1-M{\left(A\xi \right)}_{\ell }^{⊺}\right)\rangle }{\langle S_{b},1_{E\times 1}\rangle }\right)\\ & =\log\left(1-\frac{\langle S_{b},M{\left(A\xi \right)}_{\ell }^{⊺}\rangle }{\langle S_{b},1_{E\times 1}\rangle }\right)≃-\langle \frac{S_{b}}{\langle S_{b},1_{E\times 1}\rangle },M{\left(A\xi \right)}_{\ell }^{⊺}\rangle \;. \end{array}$$

*The final expression in (Equation 19) in matrix notation is (Equation 16) and (Equation 18). For dual energy CT (the case where M=B=2), the use of the inverse of U has been proposed in [29]. We emphasize that while we utilize the linearization $H^{\prime }\left[0\right]\left(\xi \right)=-A\xi U^{⊺}$ as an auxiliary tool, we actually solve the full nonlinear problem. However, the linearized LSQ problem ${∥A\xi U^{⊺}-Y_{H}∥}_{2}^{2}/2\to \min_{\xi }$ is also of interest in its own. Theoretically proven convergent algorithms for such problems can be found in [30,31].*


The derivatives in Theorem 1 have a clear composite structure that we exploit in our algorithms. This is discussed next.

**Remark** **7 (Composite structure of derivatives).**
*Consider $X\in {\left(R^{M}\right)}^{N_{x}}$ as a signal of size $N_{x}$ with M channels (each channel is a material) and data $Y_{H}\in {\left(R^{B}\right)}^{N_{y}}$ of size $N_{y}$ with B channels (each channel represents an energy bin). Then, we can write H(X)=Φ(AX), where the following nonlinear function:*

(20)
$$\Phi \left(Z\right)=\log_{N_{y}\times B}\left(\exp_{N_{y}\times E}\left(-ZM^{⊺}\right)\cdot {\underline{S}}^{⊺}\right)$$

*operates on the multichannel sinogram Z=AX along the (horizontal) channel dimension and $\underline{S}=S⊘\left(S\cdot 1_{E\times N_{y}}\right)$ are the normalized effective spectra. Application of the chain rule and some computations results in:*

(21)
$$\begin{array}{cc} H^{\prime }\left[X\right]\left(\xi \right) & =\Phi^{\prime }\left[AX\right]\left(A\xi \right) \end{array}$$


(22)
$$\begin{array}{cc} \Phi^{\prime }\left[Z\right]\left(\zeta \right) & =-\left(\exp_{N_{y}\times E}\left(-ZM^{⊺}\right)⊙\left(\zeta \cdot M^{⊺}\right)\right)\cdot S^{⊺})⊘\left(\exp_{N_{y}\times E}\left(-ZM^{⊺}\right)S^{⊺}\right) \end{array}$$


(23)
$$\begin{array}{cc} \Phi^{\prime }{\left[Z\right]}^{*}\left(\eta \right) & =-(\exp_{N_{y}\times E}\left(-ZM^{⊺}\right)⊙\left(\left(\eta ⊘\left(\exp_{N_{y}\times E}\left(-ZM^{⊺}\right)S^{⊺}\right)\right)\cdot S\right))\cdot M\;. \end{array}$$

*from which we recover (Equation 12) and (Equation 14). Furthermore, for the zero material sinogram Z=0, we get $\Phi^{\prime }\left[0\right]\left(\zeta \right)=-\zeta U^{⊺}$ and $\Phi^{\prime }{\left[0\right]}^{*}\left(\eta \right)=-\eta U$ with $U=\left(S\cdot M\right)⊘\left(S\cdot 1_{N_{y}\times M}\right)$ as in Remark 6. Equations (Equation 21)–(Equation 23) factorize the derivative $\Phi^{\prime }\left[Z\right]$ and its adjoint into two separate parts: a high-dimensional ill-posed but linear part A operating in the pixel dimension and small size well-posed but nonlinear part operating in the channel dimension. Our algorithms will target this structure for fast and effective iterative updates.*


### 3.2. Gradient and Newton—One-Step Algorithms

It is helpful to start with gradient based method for minimizing the LSQ functional (Equation 8). Our first method, CP-full, can be seen as a modified version of a hybrid between the standard gradient iteration (or Landweber’s method) and the Gauss–Newton method. Our second method, CP-fast, involves a simplification based on linearization around zero.

Gradient-based one-step algorithms for solving the MSCT problem $HX=Y_{H}$ using the LSQ functional D(X) start with the optimality condition $\nabla D\left[X\right]=H^{\prime }{\left[X\right]}^{*}\left(H\left(X\right)-Y^{H}\right)=0$ and fixed point equations derived from it. Applying a nonstationary positive-definite preconditioner $Q_{k}$ and a step size $\omega_{k}>0$ results in:(24)$$X_{k+1}=X_{k}-\omega_{k}\cdot Q_{k}H^{\prime }{\left[X_{k}\right]}^{*}\left(H\left(X_{k}\right)-Y^{H}\right)\;.$$
Explicit expressions for H and $H^{\prime }{\left[X_{k}\right]}^{*}$ are given by (Equation 7) and (Equation 14). Particular choices for the preconditioner and the step size yield various iterative solution methods. including Landweber’s iteration, the steepest descent method, Gauss–Newton iteration, Newton-CG iterations, or Quasi-Newton methods. To motivate our algorithm, it is most educational to discuss the Landweber and the Gauss–Newton iteration.
**Landweber** **method:**In the context of inverse problems, the standard gradient method with a constant step size ω is known as the (nonlinear) Landweber iteration $X_{k+1}=X_{k}-\omega \cdot H^{\prime }{\left[X_{k}\right]}^{*}\left(H\left(X_{k}\right)-Y^{H}\right)$, which is (Equation 24) for the case where $Q_{k}$ is the identity and $\omega_{k}=\omega$. Landweber’s iteration is stable, robust, and easy to implement. It is even applicable in ill-posed cases, where, with an appropriate stopping criterion, it serves as a regularization method [32]. On the other hand, it is also known to be slow in the sense that many iterative steps are required. In our case, this is due to the ill-conditioning of the forward operator.**Gauss–Newton** **method:**Several potential accelerations of Landweber’s method exist, and preconditioning seems one of the most natural ones. In the context of nonlinear least squares, the Gauss–Newton method and its variants are well-established and effective. In this case, one chooses the preconditioner $Q_{k}={\left(H^{\prime }{\left[X_{k}\right]}^{*}H^{\prime }\left[X_{k}\right]\right)}^{-1}$ in (Equation 24), which results in:
(25)$$X_{k+1}=X_{k}-\omega_{k}\cdot {\left(H^{\prime }{\left[X_{k}\right]}^{*}H^{\prime }\left[X_{k}\right]\right)}^{-1}H^{\prime }{\left[X_{k}\right]}^{*}\left(H\left(X_{k}\right)-Y^{H}\right)\;.$$The Gauss–Newton method (Equation 25) has the potential to significantly reduce the required number of iterations. On the other hand, each one of these iterations is numerically costly, as it requires inversion of the nonstationary normal operator $H^{\prime }{\left[X_{k}\right]}^{*}H^{\prime }\left[X_{k}\right]$. Moreover, due to ill conditioning, the inversion needs to be regularized [28,33,34]. The algorithms proposed in this paper use simplifications that do not need to be regularized and avoid the costly inversion of the normal operator.

### 3.3. Proposed Algorithms

Now we move on to the proposed iterative algorithms for MSCT. We start with CP-full, which is a gradient-based algorithm with channel preconditioning. We then derive CP-fast, which is a derivative-free iterative algorithm using a stationary adjoint problem.
**CP-full:** The first proposed algorithm is an instance of (Equation 24). Instead of no preconditioning, as in Landweber’s method, or the costly preconditioning $Q_{k}={\left(H^{\prime }{\left[X_{k}\right]}^{*}H^{\prime }\left[X_{k}\right]\right)}^{-1}$, as in the Gauss–Newton method, we propose preconditioning with the channel mixing term Φ only. That is, we exploit the factorization H(X)=Φ(AX) and propose the choice $Q_{k}={\left(\Phi^{\prime }{\left[AX_{k}\right]}^{*}\Phi^{\prime }\left[AX_{k}\right]\right)}^{-1}$ for the preconditioner. This results in the following CP-full iteration:
(26)$$X_{k+1}=X_{k}-\omega_{k}\cdot A^{⊺}\cdot (\Phi^{\prime }{\left[AX_{k}\right]}^{*}{\left(\Phi^{\prime }\left[AX_{k}\right]\right)}^{-1}\Phi^{\prime }{\left[AX_{k}\right]}^{*}\left(H\left(X_{k}\right)-Y^{H}\right)\;.$$While efficiently addressing the nonlinearity via a Gauss–Newton-type preconditioner in the channel dimension, it is computationally much less costly than the full Gauss–Newton update. Instead of inverting $H^{\prime }{\left[X_{k}\right]}^{*}H^{\prime }\left[X_{k}\right]$, which in matrix form has size $\left(N_{x}M\right)\times \left(N_{x}M\right)$ in the Gauss–Newton method, it requires inversion of the smaller M×M matrices $\Phi^{\prime }{\left[AX_{k}\right]}^{*}\Phi^{\prime }\left[AX_{k}\right]$ only, which can be done separately for each pixel in the projection domain. Assuming M,B=O(1) and $N_{y}=O\left(N_{x}\right)$, this dramatically reduces the cost of preconditioning from $O(N_{x}^{3})$ to $O(N_{x})$ per iterative update.**CP-fast:** In the derivative-free version, we go one step further and completely avoid the derivative $H^{\prime }\left[X_{k}\right]$. For that purpose, we replace the derivative $\Phi^{\prime }\left[AX_{k}\right]$ in (Equation 26) by the derivative at zero. According to Remark 6, we have $\Phi^{\prime }\left[0\right]\left(\zeta \right)=-\zeta U^{⊺}$ with $U=\left(S\cdot M\right)⊘\left(S\cdot 1_{E\times M}\right)$. Now, with $U^{‡}={\left(U^{⊺}U\right)}^{-1}U^{⊺}$ denoting the pseudoinverse of U, we arrive at the iterative update:
(27)$$\;X_{k+1}=X_{k}-\omega_{k}\cdot A^{⊺}\cdot \left(H\left(X_{k}\right)-Y^{H}\right)\cdot {\left(U^{‡}\right)}^{⊺}\;.$$We refer to (Equation 27) as the derivative-free fast channel-preconditioned (cp-fast) iteration. It only involves the derivative at zero, which can be computed once before the actual iteration. In this sense, it is actually derivative-free and fast. It can be interpreted as using the full nonlinear model for the forward problem, the linearization at zero for the adjoint problem, and including channel preconditioning.

Both iterations (Equation 26) and (Equation 27) are of fixed-point type and we, therefore, expect convergence for sufficiently small step sizes. Theoretically, proving convergence seems possible, but this is beyond the scope of this paper. As (Equation 26) is of gradient type, it seems easier to derive convergence for CP-full, while for the derivative free version CP-fast, such a proof seems challenging. Note further that for the results presented below, we integrated a positivity constraint by alternating iterative updates with the orthogonal projection onto the cone of nonnegative images.

## 4. Numerical Simulations

We compared our algorithms CP-full and CP-fast to existing iterative one-step algorithms in MSCT. Our evaluation builds on [11], which compares five such algorithms and provides open source code (https://github.com/SimonRit/OneStepSpectralCT, accessed on 13 July 2022.) that is used for our results. We compare CP-full and CP-fast with the best performing one of [11], and further with a two-step method.

### 4.1. Comparison Methods

The work [11] compares the following iterative one-step algorithms for MSCT in terms of memory usage and convergence speed to reach a fixed image quality threshold:Ref. [19] derives a nonlinear CG method for a weighted LSQ term;Refs. [20,22] derive surrogate approaches for Poisson maximum likelihood;Ref. [21] extends [22] by including Nesterov’s momentum acceleration;Ref. [23] generalizes the Chambolle–Pock algorithm [35] to nonconvex functionals.

Specifically, ref. [11] found the algorithm of [21] (referred to as Mechlem2018) to be significantly faster than the other four methods, and thus, we use it for comparison.

In addition, we compare with the algorithm [17] (referred to as Niu2014) as a prime example of an image domain two-step method. They use a penalized weighted least squares estimation technique applied to an empirical linear model. Note that more recently, data-driven methods based on neural networks and deep learning have also been proposed. Such methods are beyond the scope of this manuscript and we refer the interested reader to the review articles [10,36].

### 4.2. Numerical Implementation

For the presented results, we build on the Matlab code of [11], which we extend with our algorithms. In particular, we work with M=3 base materials (water, iodine, and gadolinium) and B=5 energy bins. The energy variable is discretized using E=150 uniform nodes between 0 and 150 keV. The attenuation functions and energy spectra used are shown in Figure 3. We use $N_{x}=256\times 256$ image pixels and $N_{y}=262450$ line integrals for the Radon transform. In particular the code https://github.com/SimonRit/OneStepSpectralCT (accessed on 13 July 2022) creates matrices:$M\in R^{150\times 3}$ for the base materials;$S\in R^{5\times 150}$ for the effective energy spectra;$A\in R^{N_{y}\times N_{x}}$ for the Radon transform.

After row normalizing $\underline{S}=S⊘\left(S\cdot 1_{E\times N_{y}}\right)$, we have $H\left(X\right)=\log(\exp\left(-A\cdot X\cdot M^{⊺}\right)\cdot {\underline{S}}^{⊺})$ for the MSCT forward model. Furthermore, noisy data *Y* are created with a different realistic forward model and Poisson noise added.

Besides M, S, A, H, and *Y*, we require implementations of $U^{‡}={\left(U^{*}U\right)}^{-1}U^{*}$ for CP-fast and $U_{k}^{‡}={\left(\Phi^{\prime }{\left[AX_{k}\right]}^{*}\Phi^{\prime }\left[AX_{k}\right]\right)}^{-1}\Phi^{\prime }{\left[AX_{k}\right]}^{*}$ for CP-full. Computing $U^{‡}$ is trivial and can be done in advance; $U_{k}^{‡}$ are computed in each step of CP-fast using (22) and (23).

### 4.3. Results

Reconstruction results using the proposed algorithms CP-fast (top row) and CP-full (second row) and the two comparison methods Mechlem2018 [21] (row three) and Niu2014 [17] (bottom row) can be seen in Figure 4. The phantom shown in the top row is made out of iodine (left), gadolinium (middle), and water (right). In all cases, we use noisy data and plot the iteration with minimal $\ell^{2}$-reconstruction error ${∥X_{k}-X_{★}∥}_{2}/{∥X_{★}∥}_{2}$, where X★ is the ground truth. All calculations are performed on the same standard laptop, where one iteration of CP-full takes around six seconds, one iteration of CP-fast takes around one second, and one iteration of Mechlem2018 [21] about four seconds. These times are comparable to Niu2014 [17], which takes around 5 to 8 s when using 500 and 1000 CG iterations. Figure 5 shows the evolution of the relative $\ell^{2}$-reconstruction error for various one-step methods. Note that for CP-fast, the minimum error in Iodine and Gadolinium is reached at approximately the same number of iterations, which shows efficient preconditioning and is important in application. Furthermore, we do not enforce that the sum over the three density images is one, and in the example, it indeed does not hold. The proposed algorithms turned out to be more stable than Mechlem2018 [21] and produce better results. In particular, CP-full gives the best results, while CP-fast is fastest.

Additional results for a different phantom are shown in Figure 6 and Figure 7. The concentration of iodine, gadolinium, and water has been computed as in [11].

## 5. Conclusions and Outlook

Image reconstruction in MSCT requires the solution of a nonlinear ill-posed problem. Iterative one-step methods are known to be accurate for this purpose. In this work, we propose two generic algorithms named CP-full (channel-preconditioned full gradient iteration) and CP-fast (channel-preconditioned fast iteration). Both algorithms use preconditioning in the channel dimen•sion only, which considerably accelerates the updates compared to Newton-type methods that require solving numerically costly linear inverse problems at each iteration. CP-fast replaces the derivative in the channel nonlinearity with linearization at zero, making it even more efficient. Both algorithms turn out to be fast and robust.

There are several future directions emerging from our work. First, proving the convergence of the two algorithms and demonstrating their regularization properties is important. Second, we will combine them with more realistic noise priors, such as Poisson noise, resulting in the maximum likelihood estimation (MLE) functional. Additionally, we will integrate explicit image priors, use plug-and-play strategies, and incorporate learned components.

## Figures and Tables

**Figure 1 jimaging-10-00098-f001:**
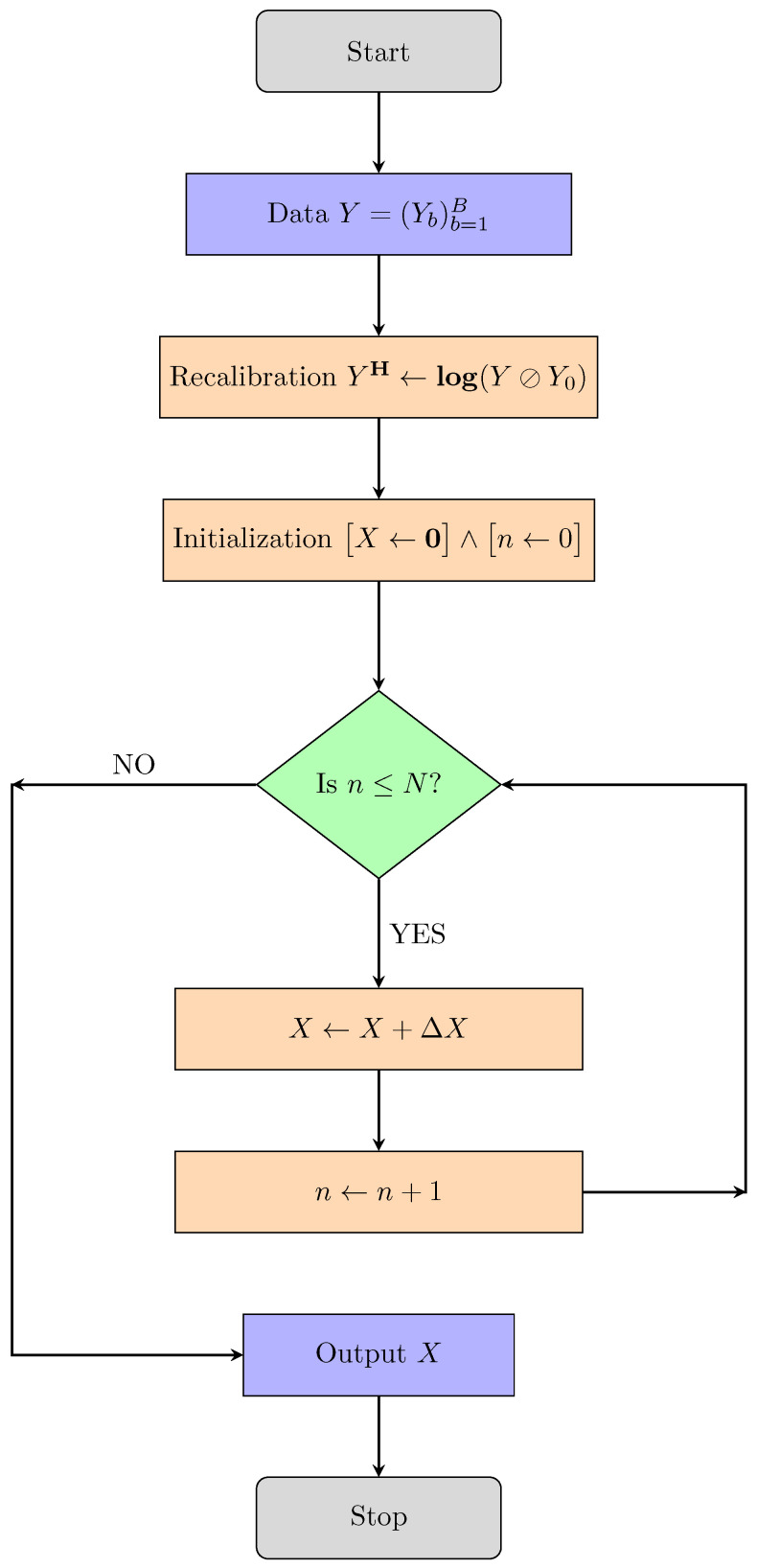
Flowchart of the structure of the algorithms proposed in this work. Specifically, we introduce CP-full and CP-fast, which differ in the specific form of the update ΔX (see Section 3.3). Details on the recalibration step are given in Remark 1.

**Figure 2 jimaging-10-00098-f002:**
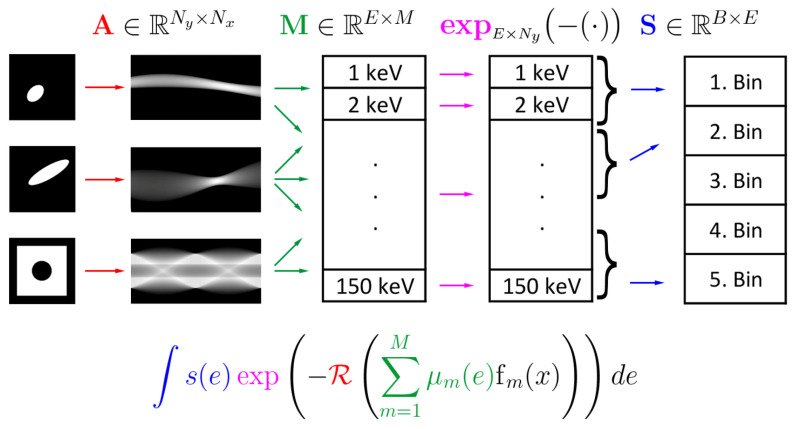
Illustration of the forward model in MSCT for m=3 materials, B=5 energy bins, and using E=150 values for energy discretization. First, the Radon transform is applied separately to each of the given material densities $X_{1}$, $X_{2}$, and $X_{3}$, resulting in three material sinograms, which can be seen as a three-channel sinogram. Next, the matrix M is applied to each pixel, resulting in 150 energy sinograms. To each of these sinograms, x↦exp(−x) is applied pointwise, resulting in 150 virtual energy data maps. By applying the matrix S pixel by pixel, one obtains the final data consisting of data maps. The continuous forward model can be visualized in a similar way by replacing the material images with continuous counterparts and the 150 energy channels with a function-valued channel.

**Figure 3 jimaging-10-00098-f003:**
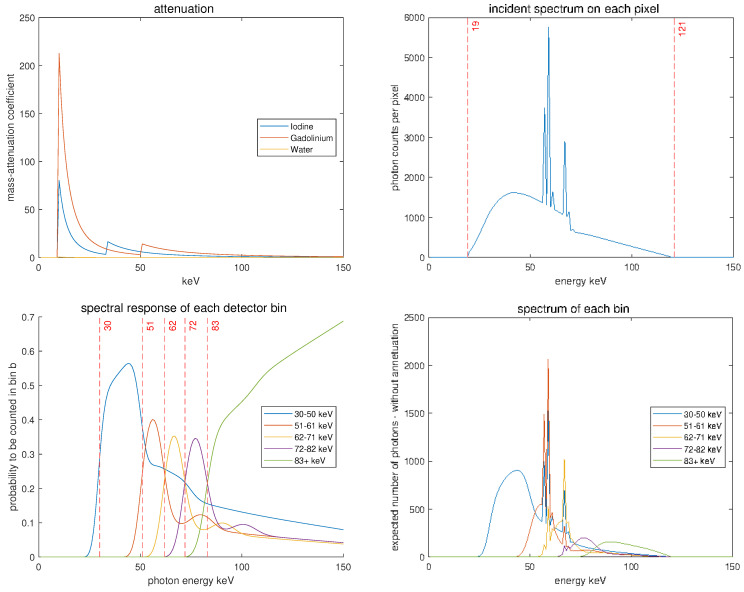
Physical parameters determining the forward model. (**Top left**): Attenuation functions. (**Top right**): Incident spectrum. (**Bottom left**): Spectral response of the detectors. (**Bottom right**): Effective spectra. The figures are based on code (modified) and data from [11].

**Figure 4 jimaging-10-00098-f004:**
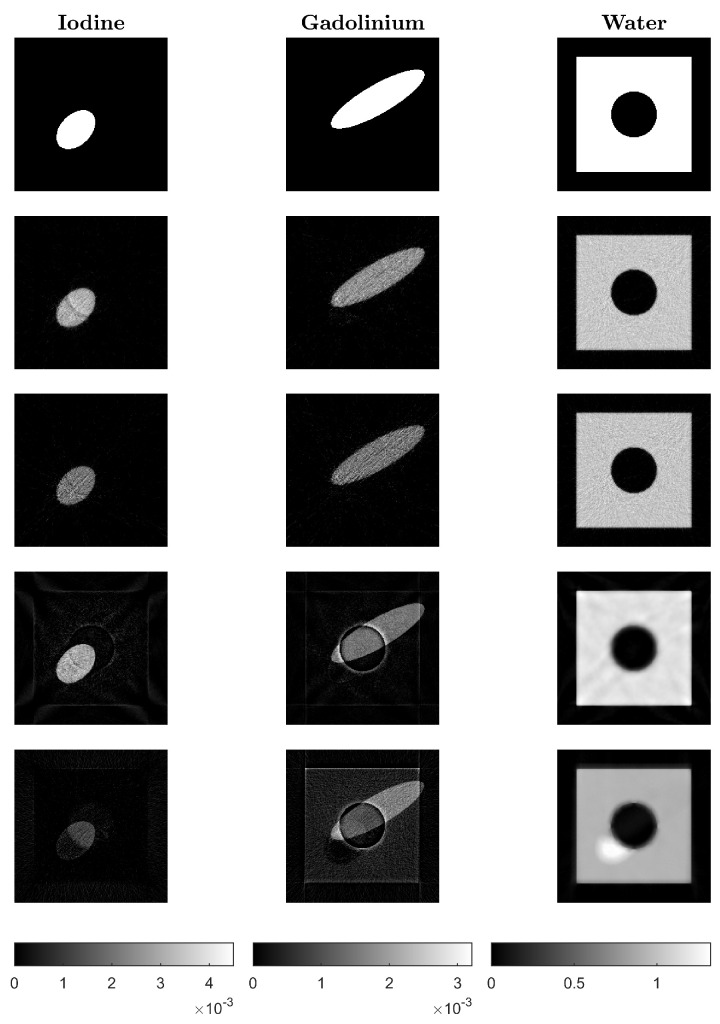
Ground truth phantom (**top row**) and reconstructions using CP-fast (**second row**), CP-full (**third row**), Mechlem2018 [21] (**fourth row**), and the two-step algorithm Niu2014 [17] (**bottom**).

**Figure 5 jimaging-10-00098-f005:**
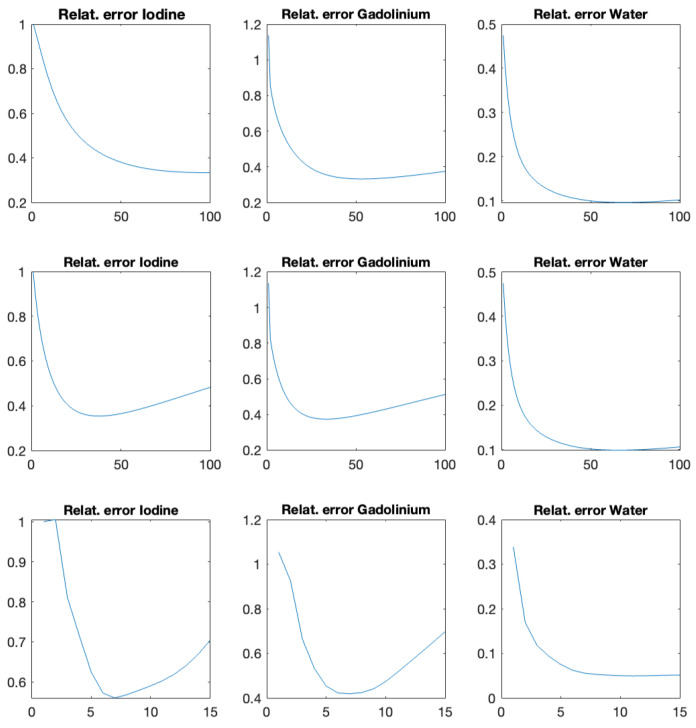
Relative reconstruction error using the proposed CP-fast (**top**), proposed CP-full (**middle**), and Mechlem2018 [21] (**bottom**) as a function of the iteration index.

**Figure 6 jimaging-10-00098-f006:**
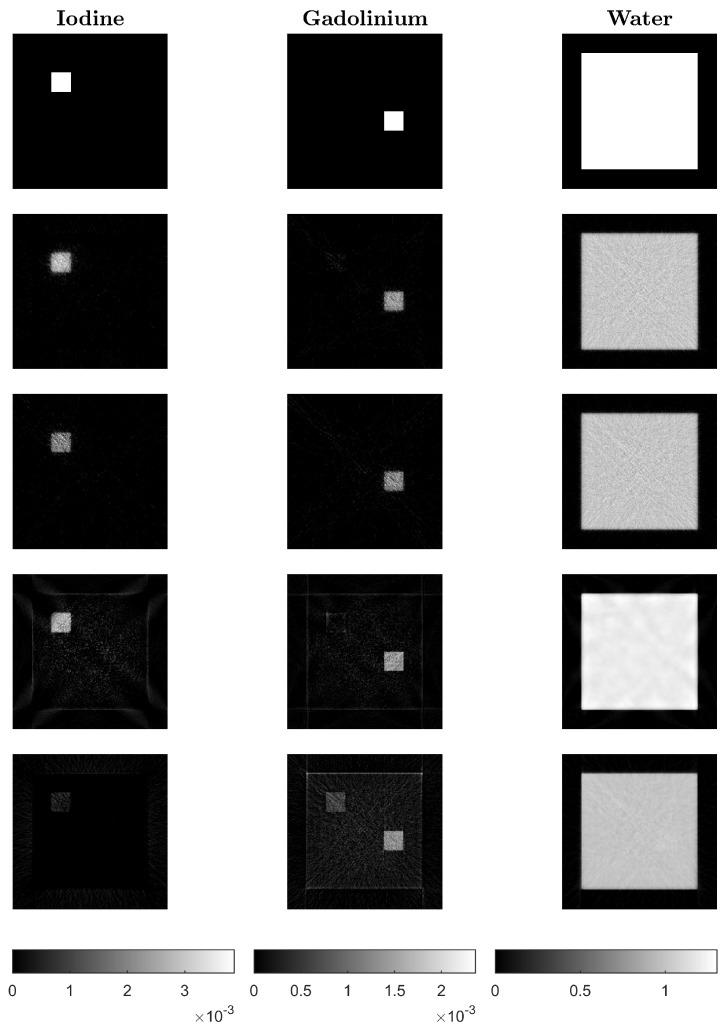
Reconstructed slices of the second phantom: Ground truth phantom (**top row**) and reconstructions using CP-fast (**second row**), CP-full (**third row**), Mechlem2018 [21] (**fourth row**), and the two-step algorithm by Niu2014 [17] (**bottom row**).

**Figure 7 jimaging-10-00098-f007:**
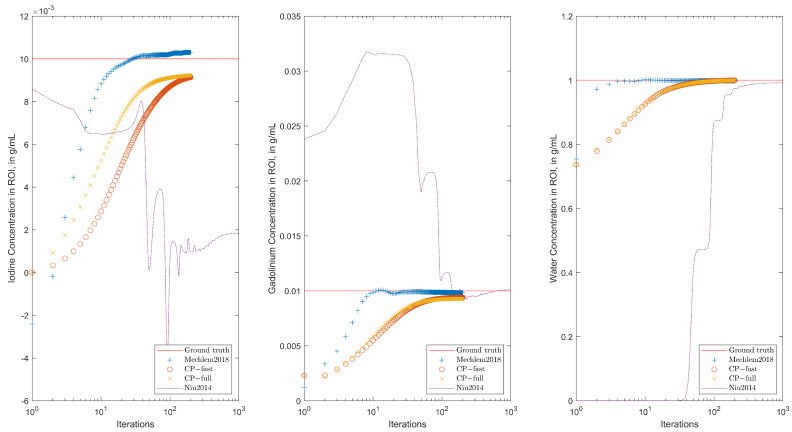
Concentration of iodine (**left**), gadolinium (**middle**), and water (**right**) for the phantom shown in Figure 6 during the iteration.

## Data Availability

No new data were created or analyzed in this study. Data sharing is not applicable to this article.

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
