# Peer review of "Derivative-Free Iterative One-Step Reconstruction for Multispectral CT"

_2313-433X, 2024, doi:10.3390/jimaging10050098_

Round 1

Reviewer 1 Report

Comments and Suggestions for Authors

Image reconstruction in multispectral computed tomography (MSCT) requires solving challenging nonlinear inverse problems, which are usually solved by iterative optimization algorithms. Existing methods require computing the derivatives of the forward mapping and possibly its regularized inverse. In this work, the authors present a simple and efficient algorithm for simple and efficient MSCT image reconstruction. The method proposed by the authors seems to be very effective. However, there are still some problems as follows:

In the first introductory part, it is suggested that it is not necessary to write the Eq;

In the second part, the theoretical part, it is suggested to give the flowchart of the algorithm so that the reader can better understand the main idea of the article;

Third, in the body part, there is too much mathematical reasoning, which is suggested to be put into the annex or appendix to strengthen the main topic;

Fourth, the comparison experiment is too little to be convincing, and it is suggested to add a side-by-side comparison.

To summarize, there is still much room for improvement in this article, and it is recommended that the author carefully revise it to reach the publication level.

Comments on the Quality of English Language

There's still a room for improvement, English.

Reviewer 2 Report

Comments and Suggestions for Authors

This is an interesting paper that presents a multispectral CT reconstruction algorithm. In particular, the proposed algorithm incorporates the advantages of both one-step and two-step techniques.

The article is well-written and transparent in its purpose. However, some observations need to be clarified.

In equation (2.2), the value of Y(s,e) equals the integral on the variable e. Please clarify why it depends on e.

The value N_x appears on line 119 of page 3. Is it possible to have an introduction of this variable?

On line 122 of page 4, the value E appears twice without any introduction.

On line 171 of page 5, the authors state "Both methods greatly reduce the number of iterations compared....". Do you have experimental data that proves this?

In the caption of Figure 4.2, next to "two-step algorithm", I would insert "Niu2014".

What are the computation times of Niu2014?

On line 359 of page 11, the authors state, "... which considerably accelerates the updates compared...". What data do you have to say this?

Reviewer 3 Report

Comments and Suggestions for Authors

I suggest that the authors provide the main contributions (1.3 section) in bullet points. 

The theory behind it is well-explained, along with the proposed modifications of the algorithm. I strongly suggest that the authors provide a pseudocode of the proposed algorithm. Is the final version of the code somewhere published? 

The paper needs more discussion and a more in-depth analysis of the results. A basic statistical comparison would also be beneficial. 

Used references are relevant and up-to-date.

Otherwise, the paper is very well-prepared. 

Comments on the Quality of English Language

Minor editing is needed.

Round 2

Reviewer 1 Report

Comments and Suggestions for Authors

  1. The response letter is drafted with excessive succinctness, necessitating the reviewer to retrieve detailed information from the revised manuscript. It is advisable for authors to infuse the correspondence letter with specifics of the amendments made, thereby facilitating a more straightforward assessment by the reviewers regarding the modifications to the manuscript;
  2. A perplexing oversight is evident wherein the revision segment of the main text remains unaltered despite the critiques posited by the reviewers. The concerns raised during the initial round of evaluation do not exhibit corresponding alterations in the revised manuscript. In essence, this renders it virtually impossible to alleviate the reviewers' reservations, and it is strongly recommended that authors engage in a thorough and meticulous revision process.

Comments on the Quality of English Language

There is still room for improving English.

Round 3

Reviewer 1 Report

Comments and Suggestions for Authors

In response to the constructive feedback from reviewers, the authors have implemented substantial revisions that thoroughly address the concerns raised. The current status of the manuscript has basically reached a publishable state and is recommended for publication in your journal.

Comments on the Quality of English Language

There is still room for improving English.

Author Response

We thank the reviewer for the positive feedback. We have again carefully read the manuscript for English formulations and have optimized some text passages. These changes are marked in orange color in the latest version.